# High Harmonic Generation Driven by Counter-Rotating Bicircular Laser Fields from Polar Chemical Bonds in *h*-BN

**Haocheng Lu [1,2,*] and Fangshu Li [1,3]**

1 State Key Laboratory of High Field Laser Physics and CAS Center for Excellence in Ultra-Intense Laser Science, Shanghai Institute of Optics and Fine Mechanics, Chinese Academy of Sciences, Shanghai 201800, China
2 Center of Materials Science and Optoelectronics Engineering, University of Chinese Academy of Sciences, Beijing 100049, China
3 School of Physics Science and Engineering, Tongji University, Shanghai 200092, China
* Correspondence: luhaocheng@siom.ac.cn

**Abstract:** High harmonic generation (HHG) driven by counter-rotating bicircular (CRB) pulses excitation has been observed from several solid targets, where circularly polarized harmonics are emitted. We study this process using time-dependent density functional theory (TDDFT) to calculate the crystal orientation dependence of the circularly polarized high harmonics from a monolayer *h*-BN. The resulted can be interpreted by the real space electron dynamics of electrons in polar chemical bonds. The yield of circularly polarized high harmonics (CHHs) can be optimized by controlling the direction of valence electron dynamics. Our findings pave the way for exploring the binding potential from spectrum and all-optically processing information.

**Keywords:** circularly polarized high harmonics; counter-rotating bicircular; high harmonic generation; *h*-BN crystal



## 1. Introduction

High harmonic generation (HHG) provides a coherent, bright, and tunable extreme ultraviolet (EUV) light and soft X-rays in a tabletop scale [1]. Circularly polarized (CP) radiation in EUV and soft X-rays spectral regions are of great importance in the application of analyzing the structures and electromagnetic properties of atoms and molecules [2–6]. Circularly polarized high harmonics (CHHs) can be used to synthesize CP attosecond pulse, which is an effective tool in probing the ultrafast electronic dynamics, attracting much attention recently [7,8].

CHHs can be generated not only in gaseous medium [2,4,6] but also from crystalline solids [9–12] by using counter-rotating bicircular (CRB) pulses. The CRB field is a special double circular laser pulse consisting of counter-rotating fundamental waves and their second harmonics, whose rotational symmetry is threefold. The produced HHG follows the rule of angular momentum conservation. In solids, due to the symmetry of crystals, CHHs can be used to probe surface chirality features [5].

In contrast to the bulk crystals, two-dimensional (2D) materials have special electronic and optical properties and inspired intensive studies in nonlinear optical phenomenon, such as HHG [10,13–22]. Experimental and theoretical studies have shown that the efficiency of HHG of monolayer 2D materials is higher than that of bulks [13,21]. In recent years, *h*-BN, as a two-dimensional transparent insulator with a wide indirect band gap, has attracted wide attention [12,20,22,23]. Due to its thermal, chemical and mechanical stability, *h*-BN is regarded as a potential candidate in deep ultraviolet optoelectronic devices [23]. Because of its high damage threshold, *h*-BN is also an ideal material for studying solid HHG [20].

Moreover, when a laser pulse interacts with a matter, such as gas-phase atoms, molecules, liquids, or dielectric solids, the electron density is driven by the periodic pulsed

electric field, moving in the restoring forces of the atomic environment. Based on the complexity of this interaction in nonlinear and strong field regions [14,24–27], researchers are conducting research on electronic and information processing at optical frequencies [28–32] by controlling electron motion over a range of optical field periods [33–35] and also working on reconstructing the band structure and potential of crystalline materials through all-optical nonlinear spectroscopy [36–38].

In this paper, we calculated the HHG of a single-layer *h*-BN crystal pumped by the CRB fields in the framework of time-dependent density functional theory (TDDFT). We investigated the non-perturbative HHG in the CRB laser fields as a function of crystal orientation. We found that the yield of HHG is threefold symmetry due to the special electron motion of the interaction between the laser field and the crystal, and linked the resulting electron motion to the direction and polarity of the chemical bond. We first interpreted the result as electron dynamics from polar valence bonds. We demonstrated that it is also possible to link information of chemical bonds and nonlinear optics using the symmetries of pulse and material, which helps us to obtain and regulate the yield of CHHs.

## 2. Methods

We calculated the three-dimensional TDDFT by propagating the molecular wavefunction under the time-dependent Kohn-Sham (KS) Equation (1)

$$i\frac{\partial}{\partial x}\psi_{n,k}(r,t) = \left[-\frac{1}{2}\nabla^2 + V_{eff,\sigma}(r,t)\right]\psi_{n,k}(r,t), \tag{1}$$

where $\psi_{n\sigma}$ is the Bloch state, $n$ is the band index, $k$ is a point in the first Brillouin zone (BZ). The time-dependent effective potential $V_{eff,\sigma}(r,t)$ is a function of the electron spin-density $\rho_\sigma(r,t)$, and it is written as Equation (2)

$$V_{eff,\sigma} = V_n(r,t) + V_H(r,t) + V_{xc,k}(r,t) + V_{ext}(r,t), \tag{2}$$

in which the first term $V_n$ represents the ionic potential. The Hartree potential $V_H$ describes the classical electron-electron interaction. The exchange-correlation potential $V_{xc}$ is used to treat all the correlations and nontrivial interactions between the electrons in the adiabatic approximation. The last term $V_{ext}$ describes the interaction potential with laser fields.

In the calculation, the pump pulses are elliptically polarized in the *x*–*y* plane. The counter-rotating fields of $E_1(t)$ and $E_2(t)$ are expressed as Equations (3) and (4)

$$E_1(t) = \sqrt{\frac{1}{1+\varepsilon_1^2}}E_1 f(t)\left[e_x\cos(\omega t) + \varepsilon_1 e_y\sin(\omega t)\right], \tag{3}$$

$$E_2(t) = \sqrt{\frac{1}{1+\varepsilon_2^2}}E_2 f(t)\left[e_x\cos(q\omega t) - \varepsilon_2 e_y\sin(q\omega t)\right], \tag{4}$$

where $E_{1,0}$ and $E_{2,0}$ stand for the amplitudes of the laser fields, and $E_{1,0} = E_{2,0} = 1 \times 10^{12}$ W/cm$^2$, $e_x$ and $e_y$ are the unit vectors, $\varepsilon_1$ and $\varepsilon_2$ are the ellipticity of the two pulses respectively, $\omega$ is the carrier frequency, $q$ represents the frequency ratio of the two fields. The envelopes of both pulses are taken to be Equation (5)

$$f(t) = \sin^2\left(\frac{\pi t}{T}\right), \tag{5}$$

with $T$ being the pulse width. The pulse width is taken to be 10 cycles of $T = 20\pi/\omega$. In the CRB field of this paper, $q$ equals to 2 unless otherwise specified. The laser wavelengths of the first and second pulses are $\lambda_1 = 1600$ nm and $\lambda_2 = 800$ nm respectively.

The monolayer *h*-BN consists of a lattice unit containing one boron atom (B) and one nitrogen atom (N) with a real-space spacing of 0.3 a.u. The B–N bond length was set to be

1.445 Å, which equals the parameter in experiments [39]. A 66-Bohr simulation box was used along the outgoing direction, and 3-Bohr absorption regions were added to each side of the monolayer, which avoided the reflection error in the spectral region and improved the spectral quality. The absorbing boundary was set as a complex absorbing potential [40], and the cap height was taken as $\eta = -1$ a.u. We sampled the 2D BZ using a $42 \times 42$ Monkhorst–Pack $k$-point grid. The fully relativistic Hartwigsen, Goedecker, and Hutter pseudopotentials [41] were used.

In this paper, adiabatic TDDFT was used to calculate the generation of harmonics in monolayer $h$-BN under strong pump pulses. Although the exchange-correlation (XC) functional LDA underestimates the band gap, many previous studies have proven that it can mcorrectly describe the band dispersion of VBs and CBs [42], thus correctly describing the inter-band and intra-band dynamics [12,43]. We calculated the HHG in the monolayer $h$-BN crystal excited by the CRB field. The HHG spectrum is obtained from the total time-dependent electronic current $j(r,t)$ as Equation (6)

$$\mathrm{HHG}(\omega) = \left| \mathrm{FT}\left( \frac{\partial}{\partial t} \int j(r,t)d^3r \right) \right|_2, \tag{6}$$

where FT denotes the Fourier transform.

### 3. Results and Discussion

Our analysis of the interaction between the pump pulses and the monolayer $h$-BN is based on the full electronic bands and the real crystal structure. Figure 1a shows the hexagonal honeycomb structure of a single-layer $h$-BN crystal in real space. Unlike graphene with the zero-band gap, the two atoms in the $h$-BN lattice are unequal. The band structure of the monolayer $h$-BN is shown in Figure 1b. The minimum direct band gap at $K$-point is 4.5 eV, and the band gaps at $\Gamma$ and $M$ points are about 6.5 eV and 5.5 eV, respectively. The highest VB and the lowest CB are shown in a blue solid line and a green solid line, respectively.

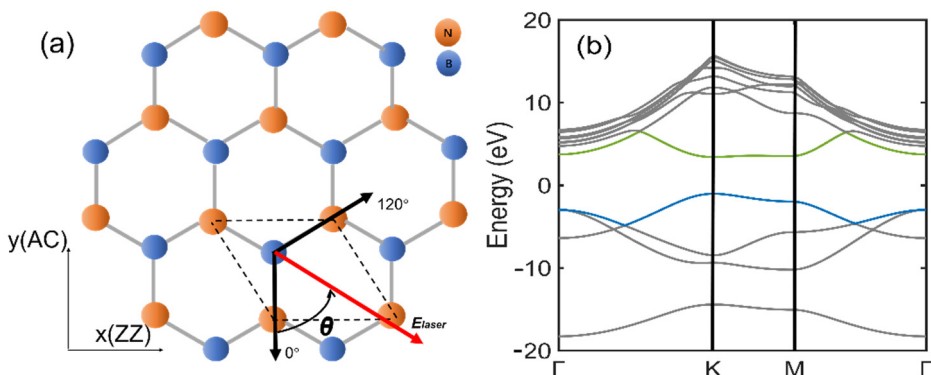

**Figure 1.** (**a**) Top-view of the crystal structure of $h$-BN. (The $x$-axis and $y$-axis are defined along the zigzag (ZZ) direction and the armchair (AC) direction, respectively. Nitrogen atoms, orange-red spheres; boron atoms, blue spheres. The red arrow is the polarization direction of the linearly polarized pump laser at an angle $\theta$ with the B–N bond. (**b**) Calculated band structure of single-layer $h$-BN. The highest VB and lowest CB are in blue and green, respectively.

Figure 2a shows the HHG spectrum of $h$-BN driven by linearly polarized pump pulse, as the laser pulse polarized along the $\Gamma$–$M$ direction. We calculated the HHG in two orthogonal directions, the $x$ and $y$-axis, respectively. The harmonic spectrum is the sum of the two orthogonal components. We can get linearly polarized HHG of all orders and the cutoff frequency can reach up to the 13th order, which agrees with the previous studies [12,22].

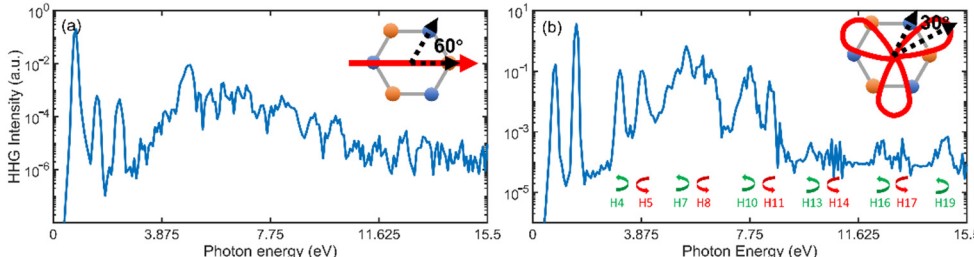

**Figure 2.** (**a**) HHG spectrum of *h*-BN in the linearly polarized pump pulse with $\theta = 60°$; (**b**) HHG spectrum of *h*-BN in the CRB pump pulse with $\theta = 30°$.

HHG from *h*-BN, however, shows several distinctive features when driven by CRB laser fields. As shown in Figure 1a, the angle $\theta$ is defined as the rotation angle of the B–N bond toward the adjacent chemical bond. In the calculation, we fixed the direction of *h*-BN and rotated the angle $\theta$ of the laser field. Figure 2b shows a typical HHG spectrum obtained at $\theta = 30°$, only several harmonic peaks with a sequence of doublets appeared, such as 4th–5th and 7th–8th, while other orders like 3th and 6th are vanishing. According to the law of conservation of energy and the law of conservation of orbital angular momentum, the allowed harmonic orders of *h*-BN are only $(3m + 1)$th and $(3m + 2)$th harmonics. Their helicity is opposite [4,6,11], the $(3m + 1)$th orders are right circularly polarized and the $(3m + 2)$th orders are left circularly polarized.

In order to study the variation of the harmonic yield with the rotation angle, we first calculated the harmonic intensities of *h*-BN crystals in linearly polarized pump pulses. The changes in harmonic yield are obtained when the *h*-BN crystal is rotated about the *z*-axis as the pump pulse polarization is fixed. Figure 3 shows the variation of harmonic yields of different orders when the wavelength of the linearly polarized pump field is 1600 nm. Since the rotational symmetry of the *h*-BN crystal is 120°, we only extracted the harmonic yield of each order as angle $\theta$ ranged from 0° to 120°. We can see that the harmonic yields of *h*-BN have two maxima within the 120° rotation. And it is centrally symmetric with 60° in the range of 120°. This indicates that the variation of the harmonic yield is 6-fold, which is consistent with the HHG from other kinds of crystals with hexagonal structures [10,22,30]. In linear polarization field, the polarization direction of laser along the B-N bond direction and N-B bond direction will produce nonlinear polarization with different intensity. However, in the multi-period laser field, the polarization direction of electrons in positive and negative periods is opposite due to the effect of the polarization direction of electric field [33,44]. Therefore, the harmonic intensity is the same for every 60° rotation of the laser field in *h*-BN.

Figure 4 shows the variation of harmonic yields of each order of *h*-BN crystals in the CRB field. There is only one maximum harmonic yield when the *h*-BN crystal is rotated by 120°, which is different from the result of the harmonic yield in Figure 3. This is consistent with the result in [5,10]. In the CRB laser field, when a hexagonal crystal with triple symmetry is rotated 360°, the harmonic yield is triple symmetric. The results in Figures 3 and 4 demonstrate the least common multiple (LCM) rule [10], i.e., the symmetry of the harmonic yield maxima of the crystal rotating for one cycle determined by the least common multiple of the symmetry of the material and the laser pulse. The symmetries of the monolayer *h*-BN and CRB fields are both $C_3$ symmetric, and their least common multiple is 3. Thus, the harmonic yield variation for one rotation is $C_3$ symmetric.

Furthermore, we find that the maximum harmonic yields of each order in Figure 4 all occur at $\theta = 60°$ and the minimum at 0°. Figure 5a shows the HHG spectra of *h*-BN crystal rotating at $\theta = 0°$ and 60° in a single-cycle CRB laser field with a fundamental frequency of 1600 nm. We can see from Figure 5a that in the single-cycle field, the harmonic spectrum still has an obvious peak shape. The reason can be attributed to the electrons are excited three times with different directions in the photoperiod in the CRB field. In addition, the harmonic yield at $\theta = 0°$ is nearly an order of magnitude smaller than that at $\theta = 60°$.

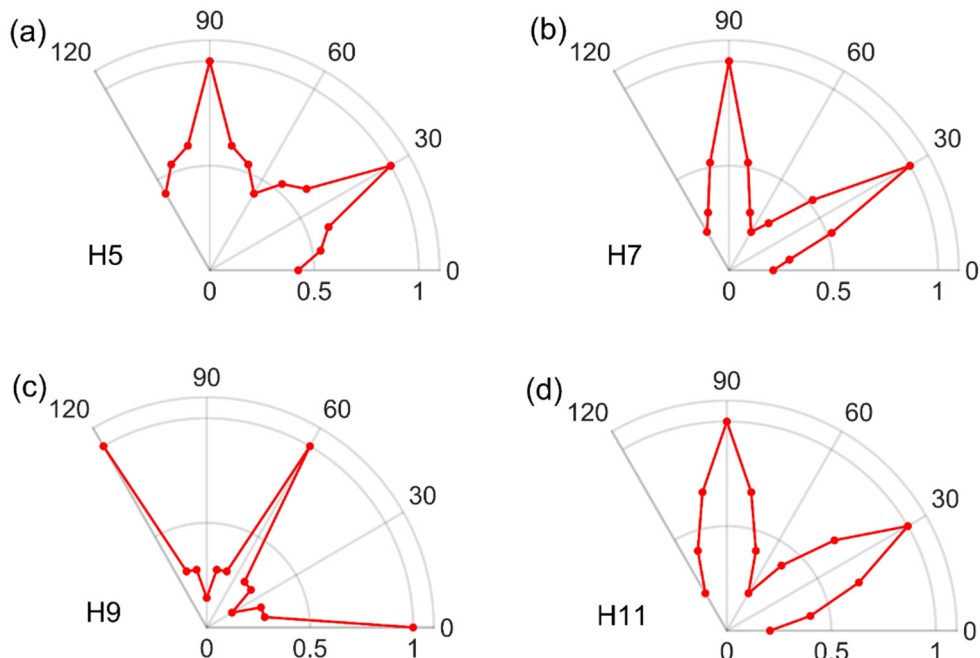

**Figure 3.** Harmonic yield as a function of linearly polarized laser fields. (**a–d**) Are the harmonic yield of H5, H7, H9, and H11, respectively.

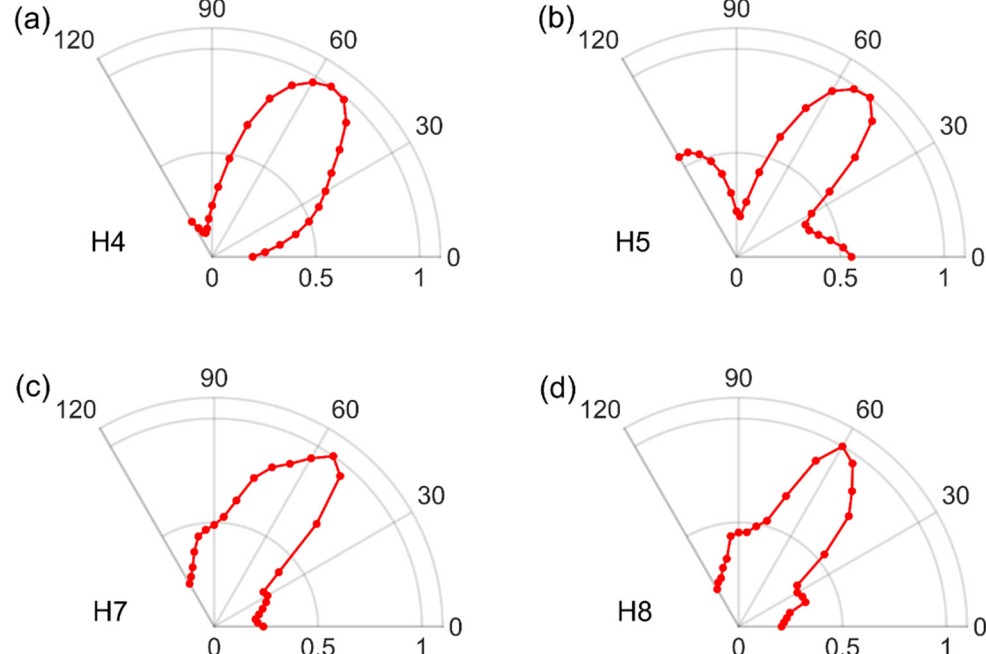

**Figure 4.** The harmonic yield variation of the CRB field is composed of a fundamental field of 1600 nm and a second harmonic field of 800 nm. (**a–d**) Are the changes in the yield of H4, H5, H7, and H8, respectively. The harmonic yields are normalized respective to each order.

In order to further study the change of harmonic intensity near the minimum bandgap energy, as shown in Figure 5b, we calculated the HHG spectra of *h*-BN crystal at $\theta = 0°$ and 60° in the single-cycle CRB laser field with the fundamental frequency of 3000 nm. The phenomenon in Figure 5b is similar to that in Figure 5a, which proves that the LCM rule is independent of the laser field wavelength. In the actual calculation, we use the trapezoidal envelope, which includes the rising edge of half a cycle and the falling edge of half a cycle. Because of the weak laser intensity, high harmonics are hardly generated.

The middle period keeps the peak intensity constant, which excites the electron during the whole photoperiod, and corresponds to the generation of most of the HHG, which simulates the generation of HHG in a single period laser field. Therefore, the enhancement of the harmonic yield can be achieved simply by rotating the crystal or the laser field.

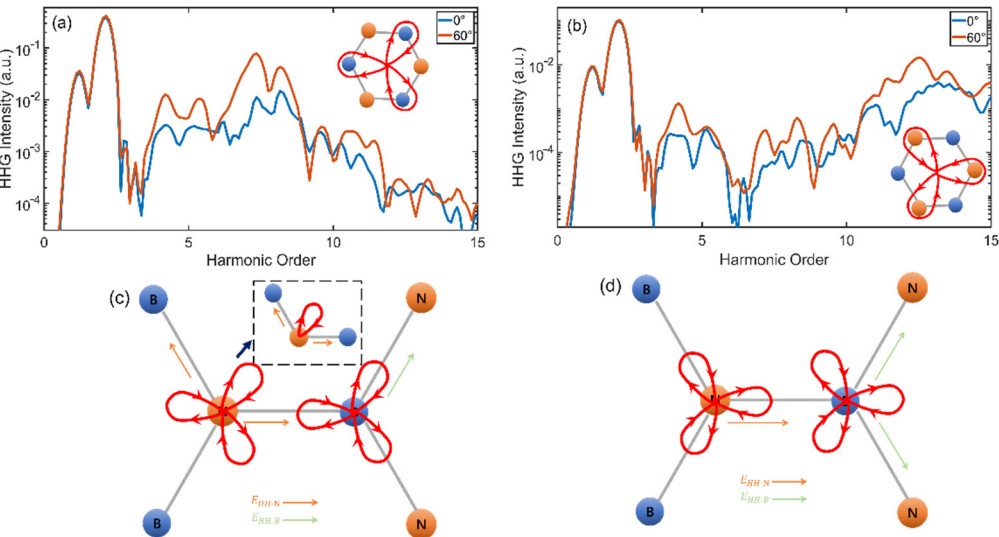

**Figure 5.** (**a**) HHG harmonic spectra of the *h*-BN crystal with $\theta = 0°$ and 60° in the CRB laser field with the fundamental frequency of 1600 nm, and the pulse duration is 6 fs (a single cycle of the fundamental field). (**b**) HHG harmonic spectra of the *m*-BN crystal with $\theta = 0°$ and 60° in the CRB laser field with the fundamental frequency of 3000 nm, and the pulse duration is 10 fs. (Insets: Diagrams of CRB laser field of *h*-BN at 0° and 60° in real space). Schematic diagrams of the process for generating high harmonics in a single layer *h*-BN crystal driven by the CRB laser field with orientation angles (**c**) $\theta = 0°$ and (**d**) 60°. Arrows indicate the direction of harmonics generated along chemical bonds. Orange arrows represent harmonics generated by electrons near N atoms, and green arrows represent harmonics generated by electrons near B atoms.

Figure 5c,d schematically depict the generation of high harmonics in a CRB laser field by electron motion in a single-layer *h*-BN crystal. In solids, high harmonics are generated along the direction of chemical bonds, although the overall direction of the total high harmonics is not necessarily along the direction of chemical bonds [30,45].

As shown in Figure 5, the Lissajous figure of the CRB field in one optical cycle is a clover shape. Each lobe is independent and equivalent, and the electrons are excited three times in one optical cycle at different angles. However, for *h*-BN, when the laser field rotates by any angle, the angle of each lobe with respect to the B–N bond is fixed, as shown in the insets in Figure 5a,b. We can analysis about the generation of the high harmonics in just one lobe, i.e., one-third of the period. When only one lobe of the laser field is considered, the vector of the electric field is projected onto three adjacent chemical bonds, producing HHG in the direction of the three bonds. The spatial directions of atomic bonds define the preferred directions for electrons motion also in momentum space, which can twist the polarization direction if the exciting field is not aligned with these preferred directions. But this can be represented as a linear superposition of the bond directions [44]. As shown in Figure 5c, at 0°, two N-B bonds and another B-N bond are excited, and excited separately in the direction of B-N bond. As shown in Figure 5d, at 60°, two B-N bonds and another N-B bond are excited, and excited separately in the direction of N-B bond.

The total yield of high harmonics $E_{HH}$ is defined as the vector sum of the harmonic yields of *x* and *y* components. As shown in Figure 5b, $\theta = 60°$, $E_N$ is the HHG generated when nonlinear polarization occurs along the N-B bond in this direction. Similarly, when $\theta = 0°$, $E_B$ is the high-order harmonic generation generated when nonlinear polarization

occurs along the B-N bond. The high harmonic generation ($\theta = 0°$) along each bond can be calculated involving its component intensities in the $x$ and $y$-axis as

$$E_{HH,0°} = \left( \cos^2 60° E_N - \cos^3 60° E_N + \cos^2 60° E_B \right) e_y + \left( \sin 60° \cos^2 60° E_N + \sin^2 60° E_B \right) e_x = \frac{3}{16} E_N + E_B \quad (7)$$

where $e_x$ and $e_y$ are the direction vectors of the $x$ and $y$-axis, respectively. When $\theta = 60°$

$$E_{HH,60°} = \frac{3}{16} E_B + E_N. \quad (8)$$

It is worth noting that no definite harmonic intensity can be given when the direction of electron polarization is distorted. Therefore, Equations (7) and (8) are only qualitative formulas.

Additionally, for any angle ranging from 0 to 120°, $E_{HH}$ can also be represented as the vector sum of the $x$ and $y$ components. In such a circumstance, because the polarization direction of the laser field is not perpendicular to or opposite to the direction of the existing chemical bond, the electrons will be nonlinear polarized on the same chemical bond.

The above-mentioned phenomenon is very similar to the suppression of harmonics in single-cycle linearly polarized pulses [44]. In each period of the CRB field, the electron motion can be regarded as three repeated motions, but in different directions. Then we can focus on the electron's dynamics in real space along one chemical bond under pulsed excitation. In the restoring force of the asymmetric atomic potential, carrier dynamics along the B–N bond direction can be described as moving in a single generalized coordinate system. This generalized coordinate moves in one dimension on the effective potential $U(\tilde{x})$. The effective potential $U(\tilde{x}) = \sum\limits_{n=2}^{\infty} c_n \tilde{x}^n$, where the classical point charge –e (elementary charge) and mass of $m_2$ (electron rest mass), and the generalized coordinate $\tilde{x}$ is the displacement from the equilibrium position. When $n = 2$, it is a linear optical response, and when $n > 2$, it expresses the nonlinear motion of electrons and can describe the harmonic intensity in Figure 5 [12]. The rotations of 0° and 60° in the CRB pulse correspond to the movement of electrons from low-potential energy atoms to equilibrium positions and from high-potential energy atoms to equilibrium positions, respectively. Electronic motion starting from low potential energies suppresses the emission of harmonics [44]. Therefore, we can obtain that the strength of $E_N$ is greater than that of $E_B$, that is, $E_{HH,60°} > E_{HH,0°}$.

The TDDFT we used in our calculations is a quantum model using electron density. The calculation results have demonstrated the difference in harmonic yield and waveform of the harmonic spectrum when rotating *h*-BN, we can also use the electron density to intuitively explain the generation of these phenomena in Figure 5a,b. The corresponding ground-state electron densities are depicted in Figure 6a. The electrons are located around N atom, where the potential is asymmetric and less steep towards the direction of the neighboring B atom. It confirms that the results of the simulations are robust against substantial variations of the parameters, as long as the system breaks the inversion symmetry and the electrons are populated mostly around N atom. The results are also robust against the relaxation and dephasing effects [44].

In the solid state, the electron density changes are fairly small under the action of the laser field [12,21,43,44], which consistent with our calculation method using adiabatic approximation [46]. Figure 6b,c show the electron densities at the peak position of the laser field minus the electron density in the ground state when the *h*-BN rotates at 0° and 60°, respectively. When the rotation angle is 0°, the electron density near the N atom increases, and the electron density near the B atom decreases. When the rotation angle is 60°, the above-mentioned phenomenon is reversed. By comparing Figure 6b,c, it can be seen that the electron density near the N atom changes larger than that near the B atom. This proves that since the potential energy of the N atom is larger than that of the B atom, the generation of high harmonics is mainly attributed to the electron motion and electron excitation near the N atom. The above discussions prove that $E_N \gg E_B$, $E_{HH,60°} > E_{HH,0°}$.

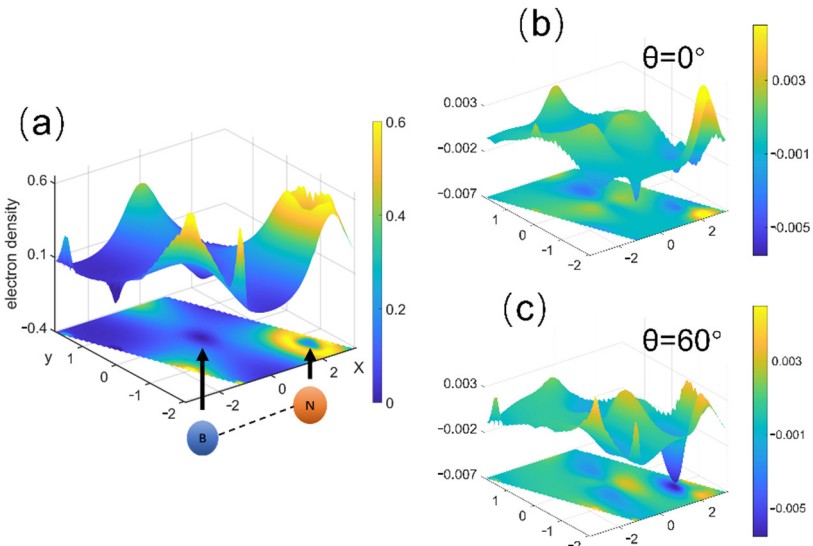

**Figure 6.** (**a**) Electron density of *h*-BN in the smallest unit cell at the initial equilibrium position. (**b**) Is the change of the excited state electron density relative to the ground state electron density at 0°. (**c**) Is the change in electron density at 60°.

We conclude that the direction, orientation, and different electron affinities of the chemical bonds determine the sign, magnitude, and shape of the nonlinear optical response induced by a single primary electron motion. Nonlinear optics in complex materials is influenced by the quantum mechanical motion of valence electrons in the potential formed by the atomic environment and the bonding structure. In *h*-BN, N atoms have greater potential energy and higher electron affinities than B atoms, and valence electrons motion around N in a time-dependent nonlinear manner. When the CRB laser field rotates at different angles, electrons are driven toward different atoms by the laser field, resulting in different harmonic intensities. When the valence electrons are driven to the shallower side of the potential (toward the B atom), their nonlinearity is enhanced, i.e., producing higher harmonic intensities. When the valence electrons are driven to the steeper side (away from the B atom), their nonlinearity is weaker, and the high harmonic intensities produced are thus weaker. Therefore, the special symmetrical structure of the CRB field affects the harmonic yield in different directions.

In the same way, we can interpret the LCM rules [10]. When we use the fundamental frequency pulse and third harmonic pulse to construct the CRB pulse as shown in Figure 7, the change of harmonic yield should be C12 rotational symmetry according to the LCM rule. Figure 7a,b are schematic diagrams of the electronic motion at $\theta$ = 0° and 30°, respectively. Considering a complete cycle of pulse, $E_{HH,0°} = E_{HH,30°} = \left(2 + \sqrt{3}\right)(E_N + E_B)$. Figure 7c shows the HHG spectra of *h*-BN crystals at rotation angles of 0° and 30° in the CRB laser field with fundamental frequency of 3000 nm and triple frequency of 1000 nm, the pulse duration is ten cycles of the fundamental field. The harmonic intensities of the two angles are comparable, proving the correctness of the LCM rule.

Next, two calculations are designed to further verify our analysis in theory. First, we calculated HHG spectra at different angles when changing the peak intensity. Figure 8a shows the HHG spectra at $\theta$ = 0° and 30° when the pulse peak intensity is changed to $5 \times 10^{11}$ W/cm$^2$. Compared with Figure 5a,b, there is no difference in spectral shape except that the harmonic intensity is two orders of magnitude smaller. If the field-driven electronic motion along the chemical bonds is dictated by the potential and waveform, the shape of spectra and the photon-order interferences should not strongly depend on the absolute electric field strength.

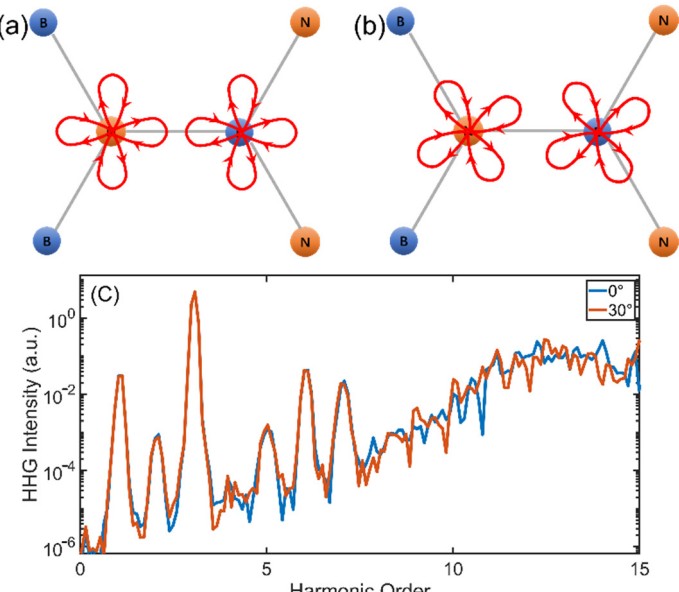

**Figure 7.** Control of electron dynamics in polar valence bonds by using both fundamental frequency pulse and the third harmonic pulse. In the CRB field composed of the fundamental frequency pulse and the third harmonic pulse, the electrons of *h*-BN move in the pulse oscillation. Schematic diagrams when (**a**) $\theta = 0°$ and (**b**) 30°. (**c**) The HHG spectra of *h*-BN crystals with rotation angles of $\theta = 0°$ and 30°.

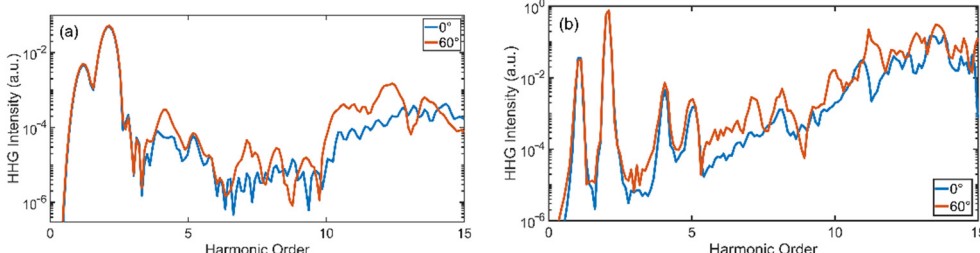

**Figure 8.** HHG spectra of the CRB field consisting of a fundamental frequency optical pulse of 3000 nm and a second harmonic pulse of 1500 nm. (**a**) HHG spectra with a duration of nearly a single cycle and a peak intensity of $5 \times 10^{11}$ W/cm$^2$. (**b**) The HHG spectra with a duration of 10 cycles and a peak intensity of $1 \times 10^{12}$ W/cm$^2$. The blue solid line is the harmonic spectra for $\theta = 0°$, and the red solid line is the harmonic spectra for $\theta = 60°$.

Finally, according to our demonstration, unlike [44], the shape of the harmonic spectra can still be quite different when the pulse is multi-periodic. Figure 8b is the harmonic spectra at $\theta = 0°$ and 60° when the pulse duration is changed to 10 cycles. The difference and spectral pattern of the harmonic intensities do not change compared to the single cycle. That is due to the positive and negative half-cycles of the linearly polarized pulse will reverse the electron movement, and the direction of the electron movement in each cycle of the CRB field is consistent with the chemical bond angle.

## 4. Conclusions

By studying the dependence of harmonic yield of *h*-BN in CRB laser field using TDDFT, we proved the LCM rule of harmonic intensity and laser field symmetry, and we found that the CHHs can be optimized by rotating the crystal. It is proved that the harmonic spectral shapes and intensities are determined by field-driven valence electron dynamics and, for heteronuclear materials, by the orientation, angle, and polarity of chemical bonds. Similarly, this contributes to the generation of high intensity CP attosecond pulses and is of great significance for the study of chiral-sensitive substances. This work also proves

the modulation of high harmonics depending on both crystal and dynamical symmetries, which can provide a reference for the study of solid HHG in the future.

**Author Contributions:** Conceptualization, H.L.; methodology, H.L.; validation, H.L. and F.L.; formal analysis, H.L.; investigation, H.L.; data curation, H.L.; writing—original draft preparation, H.L.; writing—review and editing, F.L. All authors have read and agreed to the published version of the manuscript.

**Funding:** This work was supported by the National Natural Science Foundation of China (12174412, 11874373), Youth Innovation Promotion Association of the Chinese Academy of Sciences (2021241), and the Scientific Instrument Developing Project of the Chinese Academy of Sciences (YJKYYQ20180023).

**Institutional Review Board Statement:** Not applicable.

**Informed Consent Statement:** Not applicable.

**Data Availability Statement:** Not applicable.

**Conflicts of Interest:** The authors declare no conflict of interest.

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
