# Peer review of "High Harmonic Generation Driven by Counter-Rotating Bicircular Laser Fields from Polar Chemical Bonds in h-BN"

_photonics, doi:10.3390/photonics9100731_

Round 1

Reviewer 1 Report

In this manuscript, using first-principles simulations, the authors study the orientation dependence of circularly polarized harmonics, and present the interpretation by the real-space electron dynamics along the polar chemical bonds.

The topic about the HHG from solids driven by counter-rotating bicircular laser fields may be interesting, but in this manuscript the strong-field community cannot get deeply new understandings about the solid-state HHG mechanisms. I do not recommend the current version of paper to publish in Photonics.

I have a few comments:

a)    In Fig 1(a), there is an error in the representation of direction. The x and y axes authors specify are along armchair and zigzag directions respectively, in which their definitions should be inverse. In addition, the full names about AC and ZZ should define clear in the manuscript.

b)    In 4 page or line 131-138, the values of theta are confused in Figs. 2(a) and 2(b).

c)    In 6 page, the detail about the derivation of Eqs. (7) and (8) should be specified. Moreover, considering both theta = 0 and theta = 60 are along the high symmetry orientation, the general analysis about theta in (0, 120) should be present. Since energy dispersions are non-parabolic and anisotropic, the nonlinear current induced harmonics intensity EN and EB in Eqs. (7) and (8) could be related to the orientation of the driving field. Thus the description that how the EN and EB vary with theta and then affect the orientation dependence of EHH is needed.

d)    In Fig. 6, the analyses about electronic density are rough.

Reviewer 2 Report

This work calculates the HHG of BN in linear and CRB fields with various orientation angles, using the TDDFT. The phenomena of the results are not new, which have been studied a lot in theory and observed in experiments. I don’t need to provide references, as this manuscript has already included some of them. As the phenomenon is not new, the value of this work should be providing new insights into the underlying physics. My understanding is that this work tries to explain the different HHG intensity for \theta = 0 and 60 degrees. However, I feel the discussions in this work are not based on serious physical investigation. At least, the interpretations contain vague and misleading statements.

1. One major discussion is based on the electron trajectories in solid, for example in Fig. 5 and related text. However, I don’t see any serious analysis of the trajectories nor do I know how the trajectories are calculated. In the text, I only see it is stated that the electron oscillates with the laser. This is so simple even in gas HHG. In solids, the dynamics of the electrons are more complicated because: 1) the electrons move not only in the laser field but also in the periodic potential of the crystal, and 2) electrons in solids are much more delocalized. Looking at Fig. 5 does not help me better understand the picture the authors propose. I don’t understand what the straight orange arrows mean as the fields are not linear. The electron trajectories are indicated by straight arrows too. However, the trajectories are not straight at all in the CRB field. Why can we think the electron moves from one atom to the other atom directly along the bond (Line 199)? The manuscript also assumes that the HHG is generated when the electron return to the initial position. This is not the case either, because there are so many atoms in solids. Besides, the HHG is considered separately by two parts from B and N. However, considering the highly delocalized feature in solids, is this picture valid?

I feel that the physical picture of HHG in solids is not seriously considered. Discussions are based on a very superficial picture and hand-waving language. This is the biggest problem for this work. Of course, I don’t mean one must use the exact physical picture and approximate pictures cannot be used. However, the authors must unambiguously clarify what is assumed and how the approximate picture is obtained from an accurate picture after reasonable approximations.

2. Besides, I think the difference in the 0-degree and 60-degree configurations has been discussed in previous works too, for example, considering the valley polarization. Can the authors also comment on the possibility of other reasons for the observed phenomenon?

3. The definition of \theta ``the angle between two adjacent B-N bonds’’ is not understandable.

4. How the scalar j(r,t) is obtained is not explained.

5. Line 152, ``This change is due to…’’ It is unclear what the ``change’’ means. It is not clear how the discussed phenomenon before this sentence is due to ``the opposite movement of …’’.

6. The symmetry of the harmonic yield vs. \theta is identified by counting the peaks. This is understandable. More strictly, I think this can be determined by the symmetry of the calculated results, which can be obviously identified in Figs. 3 and 4 too.

7. Results from single-cycle laser fields are discussed. It is important to clarify how HHG under single-cycle fields is calculated. In the 2nd paragraph of page 5, it says that there is no HHG peaks for \theta=0 but the peaks are obvious for \theta=60. On the one hand, I don’t see such an obvious difference from the figure. On the other hand, the HHG peaks in general originate from interference from different optical cycles. Why can there be such an obvious difference between two different angles in the case of single-cycle fields? In the current manuscript, the authors did not provide further explanation.

8. How Eqs. 7 and 8 are obtained is not clearly explained. Related to my first question, the scalar E_N and E_B, namely harmonics generated by electrons near respective atoms, are not well defined.

9. I think the yield of HHG is more dependent on the density of the excited electron than the ground state electron. The authors have checked that there is more excitation near N (Line 247). I would suggest showing this result. In Fig. 6, why is there a hole of the density at N?

10. A better way to prove the LCM in Fig. 7 is to present the angular-dependent HHG yield as in Figs. 3 and 4. However, as this requires more calculation, this is just an optional suggestion.

In summary, I could not suggest publication of the manuscript at least in the current form. To be published, the discussions of a manuscript should be at least scientifically sound and clear.

Round 2

Reviewer 1 Report

I am pleased to confirm that your revised manuscript has been accepted for publication in the Photonics

Author Response

Thanks to the reviewer for your suggestions and questions.

Reviewer 2 Report

Dear Editor,

The authors have changed their interpretation based on a trajectory picture into a quantum picture, which is more reasonable than the trajectory one. The authors have also satisfactorily responded to my other questions. I believe the manuscript has been significantly improved. Thus, I recommend publication of this manuscript. 

Regarding the current form, I have a few minor suggestions the authors may consider:

1.     The interpretation is based on `` When the valence electrons are driven to the shallower side of the potential (toward the B atom), their nonlinearity is enhanced’’ and vice versa. Could the authors add more discussion or references to more clearly elaborate and support this statement? 

2.     The use of a rectangular envelope (Line 189) makes me (and possibly other readers) worried. Generally, a too steep switching on of the laser field would lead to a non-physical problem, even though the field strength begins with zero. In the response letter, the authors assure that there will not be any problem. Have the authors checked this? In any case, considering the possible doubt from readers, I think it is better to clarify this issue by adding some of the discussions to the manuscript.
